

# Seasonal dynamics of microbial diversity in the rhizosphere of *Ulmus pumila* L. var. *sabulosa* in a steppe desert area of Northern China

Tianyu Liang[1],[*], Guang Yang[1],[*], Yunxia Ma[1], Qingzhi Yao[2], Yuan Ma[3], Hui Ma[1], Yang Hu[1], Ying Yang[1], Shaoxiong Wang[1], Yiyong Pan[1] and Gangtie Li[1]

[1] College of Desert Control Science and Engineering, Inner Mongolia Agricultural University, Hohhot, Inner Mongolia, China
[2] College of Life Sciences, Inner Mongolia Agricultural University, Hohhot, Inner Mongolia, China
[3] Desert Forest Experimental Center, China Academy of Forestry, Bayan Nur City, Inner Mongolia, China
[*] These authors contributed equally to this work.

Corresponding author
Gangtie Li, 13848817183@163.com

## ABSTRACT

The seasonal dynamics of microbial diversity within the rhizosphere of *Ulmus pumila* L. var. *sabulosa* in the hinterland of the Otindag Sandy Land of China were investigated using high-throughput sequencing of bacterial 16S rRNA genes and fungal ITS region sequences. A significant level of bacterial and fungal diversity was observed overall, with detection of 7,676 bacterial Operational Taxonomic Units (OTUs) belonging to 40 bacteria phyla and 3,582 fungal OTUs belonging to six phyla. Proteobacteria, Actinobacteria, and Firmicutes were the dominant bacterial phyla among communities, while Ascomycota, Basidiomycota, and Zygomycota were the dominant phyla of fungal communities. Seasonal changes influenced the α-diversity and β-diversity of bacterial communities within elm rhizospheres more than for fungal communities. Inferred functional analysis of the bacterial communities identified evidence for 41 level two KEGG (Kyoto Encyclopedia of Genes and Genomes) orthology groups, while guild-based analysis of the fungal communities identified eight ecological guilds. Metabolism was the most prevalent bacterial functional group, while saprotrophs prevailed among the identified fungal ecological guilds. Soil moisture and soil nutrient content were important factors that affected the microbial community structures of elm rhizospheres across seasons. The present pilot study provides an important baseline investigation of elm rhizosphere microbial communities.

## INTRODUCTION

The Otindag Sandy Land (OSL) region is located in the southern Xilin Gol Grassland of central Inner Mongolia. The region is characterized as a typical semiarid area of northern China (*Su et al., 2009*; *Mason et al., 2009*; *Gong et al., 2013*). The OSL is one of the four

great sandy regions of China, and desertification of this region is particularly serious in the farming-pastoral ecotone of northern China (*Li et al., 2016*). The OSL is the nearest sand source to Beijing, and dust transport from OSL endangers northern China and other Asian countries including the Koreas and Japan (*Cheng et al., 2005*). Therefore, land desertification in the OSL has become a global problem that requires urgent mitigation. Vegetation plays an important role in the ecosystem of sandy lands, and this is especially true for the dominant vegetation species in these areas. Specific keystone species control the functions and structures of both communities and their sandy land environments. Consequently, the degradation or loss of dominant species is the main cause of sandy land ecosystem degradation and the expansion of desertification. Protecting the vegetation of sandy lands is thus an important ecological solution to control desertification of sandy lands.

A single species of elm grows in OSL (*Ulmus pumila* L. var. *sabulosa*) that belongs to the Ulmaceae family (*Guo, Li & Li, 1988*). The well-developed root system, lush branches and leaves, and drought and cold resistance of the species all contribute to the wide distribution of elms in nearly all kinds of sites and conditions in OSL, including in fixed and semi-fixed dunes and flat sands. The distribution of elms in the OSL is sparse and patchy, forming elm open woodland grassland landscapes across OSL (*Liu et al., 2013*; *Li et al., 2011*; *Yang et al., 2014*). Open elm woodlands are an important component of OSL ecosystems and dominate communities during sand vegetation succession (*Zhao et al., 2016*). Furthermore, sparse elm woods play a vital role in various ecosystem services, especially wind prevention and sand fixation (*Wang et al., 2015*). However, recent studies have indicated that the area occupied by open elm woodlands is decreasing and that elm seedlings renew slowly, which is primarily due to over-grazing (*Wesche et al., 2011*; *Su et al., 2014*). These issues have attracted considerable research attention. However, research has primarily focused on investigating aboveground components of sparse elm forests including distribution patterns, community structures, and seedling renewal rates (*Tang, Jiang & Wang, 2014*). In contrast, research about underground components of elm forests have mainly focused on root distributions. But there is little knowledge on the soil microorganisms within elm rhizosphere, although their diversity may have a role for the capacity of this tree species to colonize and resist in such harsh environment.

Numerous recent studies have demonstrated rich microbial community resources within the rhizospheres of tall trees, as has been observed for herbs and crops. For example, *Feng et al. (2012)* evaluated bacterial and arbuscular mycorrhizal fungal (AMF) community diversity in the rhizospheres of eight plant species in the Liudaogou watershed within the Loess Plateau of China. Both bacterial and AMF community diversity were higher in the rhizospheres of *Robinia pseudoacacia* compared to other species, which could contribute to the capacity of *R. pseudoacacia* to be used for vegetation restoration as a pioneer species in this region (*Feng et al., 2012*). Further, the diversity of rhizosphere soil microbial communities of Chinese Pine (*Pinus tabulaeus*) have been investigated in the Loess Plateau. *P. tabulaeus* is widely used for restoring degraded ecosystems. These analyses revealed a higher diversity in bacterial and fungal communities within natural secondary forests than in plantations (*Yu, Wang & Tang, 2013*), which may be important for the use of this tree

species in restoration applications. Concomitantly, numerous investigations have indicated that soil microbial communities are significantly affected by soil types and physical and chemical properties. For example, changes in soil physical and chemical properties caused by seasonal changes were the key factors associated with variation in rhizosphere bacterial communities of *Pinus* (*Wang et al., 2018*).

The rhizosphere is the narrow zone of soil that surrounds roots, is influenced by root exudates, and generally contains a high diversity of microorganisms (*Mendes et al., 2011*). Microbiota in rhizospheres are considerably more diverse than clod communities because root surfaces provide numerous microenvironments for rhizosphere microorganisms to leverage (*Bais et al., 2006*; *Jackson et al., 2012*). Within these microenvironments, microorganisms interact with root systems in complex ways (*Newton et al., 2010*). For example, rhizospheric soil microorganisms can affect the decomposition and transformation of soil nutrients, in addition to plant absorption and utilization of nutrients (*Buée et al., 2009*). In addition, rhizospheric microorganisms can promote the growth of plant roots, increase the absorption area of roots, and indirectly improve plant nutrition (*Morris et al., 2010*). Soil microbial biomass is also a crucial index to measure soil fertility and nutrition (*Rodriguez et al., 2008*). Conversely, plants influence microorganisms via their root secretions and stimulate microbiome activities by regulating biological and/or abiotic environments, thereby contributing to the formation of plant-specific microbiomes (*Tian & Gao, 2014*). However, most microorganisms are not culturable (*Schenk, Carvalhais & Kazan, 2012*) and microorganisms are highly diverse (*Berendsen, Pieterse & Bakker, 2012*). Thus, appropriate characterization and enumeration methods are key to investigating *in situ* microbial diversities.

Understanding the diversity of rhizosphere microorganisms is the basis of understanding interactions between microorganisms and plants. Few studies have investigated the microbial diversity of elm tree rhizospheres (*Mendes et al., 2014*). Consequently, analysis of bacterial and fungal rhizosphere community diversity and composition via sequencing of 16S rRNA gene and internal transcribed spacer (ITS) sequences, respectively, was conducted for elm rhizospheric samples recovered from different seasons in the hinterland of OSL. To better understand the possible functional changes associated with the seasonal variations in elm rhizosphere microbial communities, inferred functional differences were investigated using PICRUSt (Phylogenetic Investigation of Communities by Reconstruction of Unobserved States), which predicts metagenome functional content from 16S rRNA gene datasets, and FUNGuild, which parses fungal OTUs into guilds based on their taxonomic assignments. The objective of the current study was to explore the response of soil microbial community diversity to seasonal changes. Further, the study aimed to address the lack of knowledge about elm rhizospheric microbial communities, which could provide useful information for the protection and utilization of sparse elm forests as harnessing rhizospheric microbiomes is increasingly recognized as a possible lever to influence plant performance through adequate management (*Sutherland et al., 2019*). The study also may be helpful to provide an important theoretical basis for understanding the ecological and environmental construction of the OSL hinterland.
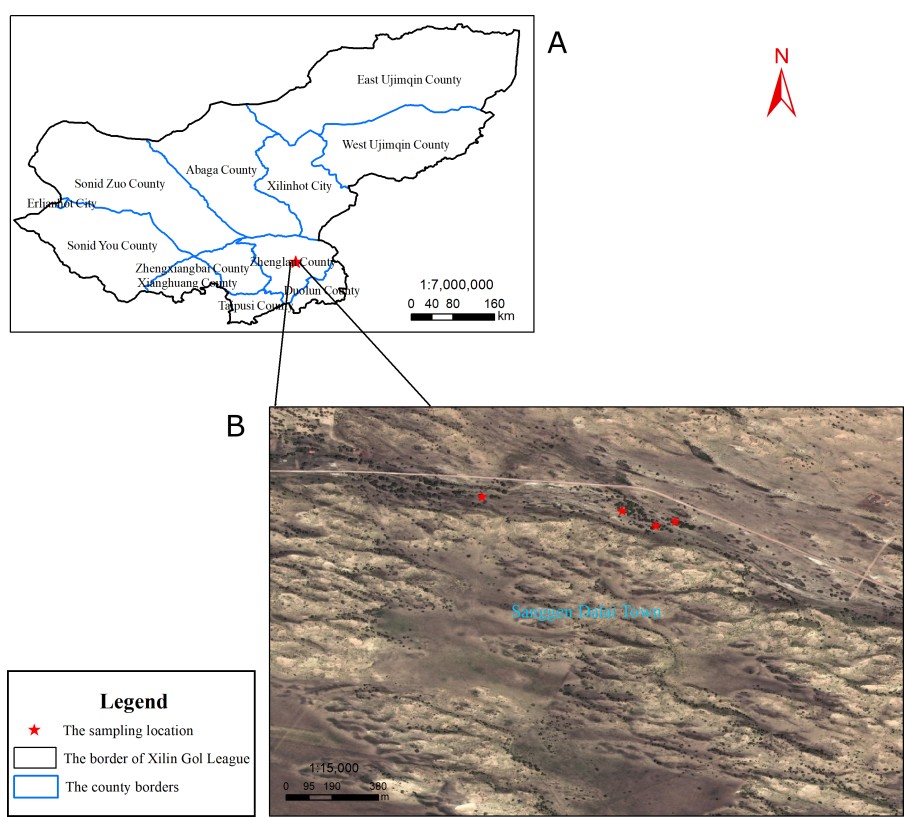

**Figure 1 Sampling locations.** (A) Xilin Gol League. (B) The specific position of sampling in Sanggen Dalai. Sanggen Dalai is located in Zhenglan County in the south of Xinlin Gol League.

# MATERIALS AND METHODS

## Sampling sites

The spatial heterogeneity of soil physical and chemical properties and soil enzyme activities are low in OSL (Table S1; Fig. S1). The study was conducted at Sanggen Dalai (latitude, 42°40′N; longitude, 115°57′E; 1,300 m AMSL), which is located in the center of OSL. The open elm woodlands of this area are the most well-preserved communities within the OSL (*Wang et al., 2015*). The area is characterized as a temperate semi-arid continental climate zone, with an annual average temperature of 1–4 °C, an average frost-free period of 105 days, annual sunshine time of 3,200 h, an annual average wind speed of 4.2 m/s, and annual precipitation of 320 mm. In this region, 90% of the dunes are fixed and semi-fixed dunes, while shifting dunes only account for a small proportion. However, increasing numbers of semi-fixed dunes have transitioned to shifting dunes due to drought and overgrazing. The zonal vegetation is that of a steppe desert. The vegetation on fixed and semi-fixed dunes is more productive, and they have been thus used as sandy pastures. Elms in the area are distributed on the windward slopes, leeward slopes, and lowlands of the sand dunes. The present study selected four lowlands with elms as fixed sampling sites (Fig. 1). Sampling on the private pasture was approved by the owner of the pasture, Ying Tao.

## Soil sample collection

Soil sampling was conducted in May (spring), August (summer), and October (autumn) of 2017. Low temperatures and thick snow cover in winter precluded soil sampling in the winter. Three individual elms were randomly selected for sampling from each of the four sampling sites (Fig. 1). At each site, the three sampled trees were 10 to 20 m apart from one another. Rhizosphere soils were collected at depths of 5–30 cm extending east, south, west, and north, 1m from the trunk of each tree. Root systems were carefully excavated using a spade, and loosely adhering soils on the roots were shaken off and discarded. The root-adherent soil particles were then collected. During each season, the soil samples from each sampling site (3 trees × 4 direction = 12 samples per site) were pooled in equal proportions to obtain a composite soil sample for each site. At each location, the same three trees were sampled at the three sampling periods. Thus, a total of 12 soil composite soil samples (i.e., 4 sites ×3 seasons) were collected. Each soil sample was subdivided into three components, with one stored at −80 °C for use in determining soil microbial community diversity, while the other was stored at 4 °C for use in soil enzyme activity determination, and the third was dried for use in determining soil physicochemical properties.

## Determination of soil physicochemical properties

Soil physiochemical properties were measured by referencing protocols described in *Soil Agricultural Chemistry analysis* (*Bao, 2000*). Total N was determined with the Kjeldah method, while available N was determined using alkaline hydrolysis diffusion. Total P was determined using NaOH-Mo-Sb colorimetry and available P was determined via NaHCO$_3$ extraction-Mo-Sb colorimetry. Available K was determined by flame photometry. Organic matter content was determined using a K$_2$Cr$_2$O$_7$ oxidation-external heating titration. pH was determined by potentiometry while moisture content was determined by measurement before and after oven drying. Lastly, soil temperature was determined using a miniature electronic temperature recorder (DS1921G; WDS). During the sampling month, continuous observation was made for one month.

## DNA extraction

An SDS (sodium dodecyl sulfate)-based method was used to extract total DNA from the rhizosphere soil samples (*Zhou, 1996*). Soil samples (5 g) were weighed and ground to a powder in a ceramic mortar using liquid nitrogen. Samples were then transferred to 50 mL centrifuge tubes. DNA extraction buffer was added (13.5 mL; 100 mM Tris–HCl [pH 8.0], 100 mM sodium EDTA [pH 8.0], 10 mM sodium phosphate [pH 8.0], 1.5 M NaCl, and 1% CTAB) in addition to 100 μL of proteinase K (10 mg/mL). The tubes were then placed on a shaker with rotation at 225 r/min for 30 min at 37 °C. Following incubation, 1.5 mL of 20% SDS was added, and the samples were incubated in a 65 °C water bath for 2 h with shaking of tubes every 15 min to ensure adequate mixing. After incubation, the mixtures were centrifuged at room temperature at 6,000 r/min for 10 min. The supernatants were then transferred to new 50 mL centrifuge tubes, while the remaining soil pellets were subjected to two additional DNA extraction rounds using 4.5 mL of DNA extraction buffer and 0.5 mL of 20% SDS, with mixing for 10 s, incubation at 65 °C for 10 min, and centrifugation as

described above. Supernatants from triplicate extractions were pooled together and mixed with an equal volume of chloroform-isoamyl alcohol (24:1). The samples were mixed by inversion and then centrifuged at 6000 r/min for 15 min. The aqueous phase was then transferred to a new 50 mL centrifuge tube. Isopropyl alcohol (0.6 × volume) was added to the samples, mixed by inversion, and allowed to precipitate at −20 °C overnight. The mixture was then centrifuged at 11,000 r/min for 20 min at 4 °C. Visible black or brown crude DNA extracts precipitated during this process. The precipitate was then transferred to a sterile 2.5 mL centrifuge tube. The crude DNA extract was washed with 70% cold ethanol, placed in a fume hood to allow ethanol volatilization, and 100 μL of sterile deionized water were finally added to dissolve DNA in water. DNA concentrations and purities were evaluated with 1% agarose gel electrophoresis.

## PCR amplification and purification

The V4 hypervariable regions of bacterial 16S rRNA genes were amplified using the forward primer 515F (5′-GTGCCAGCMGCCGCGGTAA-3′) and reverse primer 806R (5′-GTGCCAGCMGCCGCGGTAA-3′) to construct bacterial community libraries for HiSeq Illumina sequencing (*Evans et al., 2014*). Taxonomic coverage of 16S rRNA genes was evaluated in https://www.arb-silva.de/search/testprime/ (*Klindworth et al., 2013*), which yielded an overall taxonomic coverage of 92.4%. Likewise, the ITS1 region of the fungal ITS was amplified using the forward primer ITS5-1737F (5′-GGAAGTAAAAGTCGTAACAAGG-3′) and reverse primer ITS2-2043R (5′-GCTGCGTTCTTCATCGATGC -3′) (*Lu et al., 2013*). ITS1 is the common prime target for the evaluation of fungal diversity through deep sequencing, but might over estimate the fungal diversity because of its variable length (*Yang et al., 2018*). All PCR reactions were carried out in 30 μL reaction volumes with 15 μL of Phusion® High-Fidelity PCR Master Mix (New England Biolabs, Ipswich, MA, USA), 0.2 μM each of forward and reverse primers, and about 10 ng of template DNA. The following PCR procedure was used for bacterial 16S rRNA gene amplification: an initial denaturation at 98 °C for 1 min, 30 cycles of (98 °C for 10 s, 50 °C for 30 s, 72 °C for 30 s), and a final extension at 72 °C for 5 min. The PCR protocol for fungal ITS comprised an initial denaturation at 95 °C for 2 min, and then 30 cycles of (95 °C for 30 s, 55 °C for 30 s, and 72 °C for 30 s), with a final extension at 72 °C for 5 min. An equal volume of 1× loading buffer (containing SYBR green) was mixed with the PCR products and the mixtures were subject to electrophoresis on a 2% agarose gel for detection. PCR products were then mixed in equimolar ratios, and the resultant pooled PCR products were purified using a GeneJETTM Gel Extraction Kit (Thermo Scientific, Waltham, MA, USA). After purification, the PCR product pools were sequenced on the Illumina Hiseq PE250 platform.

## Bioinformatics and statistical analysis of sequence data

Samples were demultiplexed using unique barcode sequences for samples, followed by cleaving of barcode and PCR primer sequences. Paired end reads from PE250 two-terminal sequencing on the HiSeq platform were then joined using the FLASH software (V1.2.7, http://ccb.jhu.edu/software/FLASH/) (*Magoč & Salzberg, 2011*), and

the joined sequences were considered as 'Raw Tags'. The QIIME software suite (V1.9.1, http://qiime.org/scripts/split_libraries_fastq.html) (*Caporaso et al., 2010*) pipeline was used to quality control and filter raw tags to obtain 'clean tags' representing high-quality sequence reads. Clean tag sequences were compared against the Gold database (http://drive5.com/uchime/uchime_download.html) (*Haas et al., 2011*) using the UCHIME Algorithm (http://www.drive5.com/usearch/manual/uchime_algo.html) (*Edgar et al., 2011*) to remove chimeric sequences and finally obtain 'effective tags'. The Uparse software suite (V7.0.1001, http://drive5.com/uparse/) (*Edgar, 2013*) was used to cluster effective tags into operational taxonomic units (OTUs) that were defined at 97% nucleotide identities. Taxonomic classification of representative bacterial OTUs was conducted using the SILVA SSUrRNA database (http://www.arb-silva.de/) based on the Mothur algorithm (*Pruesse et al., 2007*). Likewise, the BLAST method (http://qiime.org/scripts/assign_taxonomy.html) (*Altschul, 1990*) within QIIME (V1.9.1) was used to taxonomically annotate fungal representative OTU sequences by comparison against the UNITE database (https://unite.ut.ee/) (*Johnson, Warburton & Mills, 2010*). The MUSCLE (V3.8.31, http://www.drive5.com/muscle/) (*Edgar, 2004*) aligner was used to generate multi-sequence alignments for all representative OTU sequences. Finally, the number of sequences per sample were rarefied to equivalent levels (48,262 sequences for Bacteria and 48,027 for Fungi). Subsequent α-diversity and β-diversity analyses were conducted on the rarefied diversity tables. Alpha diversity indices were calculated using QIIME (V1.9.1) and rarefaction curves were drawn using the R software package (*R Core Team, 2013*).

One-way ANOVA (analysis of variance) and Bonferroni corrections for multiple comparisons were conducted in the SPSS 20.0 software suite. QIIME (V1.9.1) was also used to calculate Unifrac distances among samples to evaluate β-diversity. The Adonis function in the vegan package for R was used to conduct PERMANOVA analysis. This is a multivariate ANOVA method that seeks to evaluate the variation between samples, as indicated by distance matrices (*Anderson, 2010*). The vegan package for R was used to generate NMDS (non-metric multi-dimensional scaling) plots (*Kruskal, 1964*) and construct RDA (redundancy discriminant analysis) biplots and evaluate the sources underlying sample variation and their relative importance via exploration of 'explanatory' and 'response' variables (*Oksanen et al., 2013*; *Kenkel et al., 2002*). Lastly, the LEfSe software package was used to conduct LEfSe analysis, using an LDA score of 4 to filter results, for identifying taxonomic groups that are consistent with experimental treatments and estimating the associated effect size through class comparisons (*Segata et al., 2011*).

## Functional analysis of bacterial and fungal communities

To predict variation in the inferred functions of bacterial communities within elm rhizospheres, PICRUSt analysis was employed. A closed-reference OTU table was first generated in QIIME and compared against the KEGG functional database to obtain predicted functions that were inferred for the bacterial communities. Individual analysis steps are outlined in the online analysis platform (http: //picrust.github.io/picrust/; *Langille et al., 2013*). FUNGuild (https://github.com/UMNFuN/FUNGuild; *Nguyen et al., 2015*)

**Table 1** Seasonal dynamics of soil physiochemical properties within elm rhizospheres of the hinterland of the Otindag Sandy Land.

|  | Spring | Summer | Autumn |
|---|---|---|---|
| Total N(mg/100g) | 79.83 ± 11.33a | 94.50 ± 17.51a | 77.64 ± 10.10a |
| Available N(mg/kg) | 91.02 ± 10.75a | 85.06 ± 9.64a | 70.06 ± 3.95a |
| Total P (mg/100g) | 10.27 ± 1.09a | 14.53 ± 1.86a | 10.46 ± 1.20a |
| Available P (mg/kg) | 3.64 ± 0.35b | 6.00 ± 0.56a | 2.17 ± 0.32b |
| Available K(mg/kg) | 451.67 ± 56.04a | 220.93 ± 29.81ab | 297.95 ± 44.29b |
| Organic matter(g/kg) | 11.72 ± 1.72a | 17.98 ± 3.59a | 16.22 ± 2.01a |
| pH | 7.40 ± 0.18a | 7.43 ± 0.14a | 7.38 ± 0.23a |
| Moisture content (%) | 5.51 ± 0.63a | 3.53 ± 0.38b | 1.27 ± 0.33c |
| Soil temperature (°C) | 15.98 ± 0.35b | 21.09 ± 0.24a | 4.42 ± 0.14c |

**Notes.**
Data indicate mean ± standard deviations.
Different letters within the same parameter indicate significant differences at the $p \leq 0.017$ level (Bonferroni corrected results).

was used to identify the potential ecological roles of fungi within the communities. To evaluate the level of difference in inferred functionalities, one-way ANOVA (analysis of variance) analysis and Bonferroni corrections for multiple comparisons were used with the inferred function dataset using the SPSS 20.0 software package (SPSS, Chicago, IL, USA).

# RESULTS

## Seasonal dynamics of soil physicochemical properties

The soils were weakly alkaline, exhibiting a pH range of 7.38–7.43 without significant differences among seasons (Table 1). The moisture content (MC) of soils over all three seasons was lower than 10%. The maximum MC was observed in spring (5.51%) and the minimum was observed in autumn (1.27%), which was significantly lower ($p < 0.017$). Surface soil (5–30 cm) temperature dynamics were consistent with those of air temperatures, wherein temperatures in summer were greater than in spring and autumn. Total P did not significantly differ among seasons, while available phosphorus levels in summer were significantly higher than those of spring and autumn ($p < 0.017$). Available K content was highest in spring and significantly differed from K contents in the autumn ($p < 0.017$). Lastly, seasonal variation in total nitrogen, effective nitrogen, and organic matter was not apparent (Table 1).

## Response of microbial community α-diversity in elm rhizosphere soils across seasons

A total of 909,174 and 945,531 16S rRNA gene and ITS sequences for bacterial and fungal communities were obtained, respectively (Table S2; Table S3). The distribution of OTUs that were defined at the 3% nucleotide dissimilarity level were then identified among samples. A total of 7,676 16S rRNA gene OTUs were observed among all samples from the three seasons. Among these, 4,933, 6,269, and 5,430 OTUs were observed in the spring, summer, and autumn samples, respectively. Of these, 3,627 OTUs were common to all samples, which accounted for 47.25% of the total OTUs (Fig. 2A). In addition, 3,582 ITS sequence OTUs were observed among all samples from the three seasons. Among these,

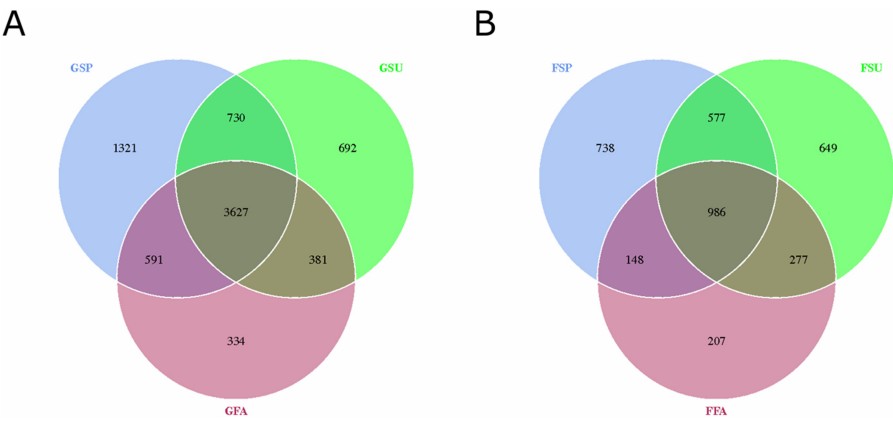

**A** GSP 730 GSU 1321 692 3627 591 381 334 GFA

**B** FSP 577 FSU 738 649 986 148 277 207 FFA

**Figure 2** **Venn diagram showing overlap in (A) bacterial and (B) fungal OTUs within elm rhizosphere communities from different seasons.** *GSP*, bacterial communities in spring samples; *GSU*, bacterial communities in summer samples; *GFA*, bacterial communities in autumn samples; *FSP*, fungal communities in spring samples; *FSU*, fungal communities in summer samples; *FFA*, fungal communities in autumn samples.

2,449, 2,489, and 1,618 OTUs were observed in the samples from spring, summer, and autumn, respectively. A total of 986 fungal ITS OTUs were common to all samples, which accounted for 27.53% of the total fungal OTUs (Fig. 2B).

After rarefying the number of sequences per sample to 48,262 and 48,027 for bacterial and fungal communities, respectively, an average of 3,579 and 1,043 OTUs were observed for the communities, respectively. The rarefaction curves and species accumulation boxplots for both bacterial and fungal communities generally reached asymptotic levels, indicating that the sample numbers and the sequencing effort applied here was adequate to capture most of the bacterial and fungal diversity in these communities (Fig. 3; Fig. S2).

Four indices of α-diversity were analyzed including the Shannon, Simpson, Chao1, and ACE. In addition, we also evaluated Good's coverage index that provides an estimate of diversity captured by the sequencing effort, which yielded estimates of 98.1% and 99.5% for the bacterial and fungal communities, respectively. The four diversity indices followed similar trends wherein bacterial diversity was highest in spring rhizospheres, followed by summer and autumn (Table 2). One-way ANOVA (analysis of variance) of the Shannon index values indicated a significant difference in diversity among the three seasons. Subsequent multiple comparisons (with Bonferroni correction) indicated that autumn communities were significantly different from those in the spring ($p < 0.017$), while the diversity of communities in the summer were not significantly different from diversity in the spring and autumn communities. In addition, one-way ANOVA revealed significant differences for Simpson indices among soils from the three seasons. Subsequent multiple comparisons indicated that the spring community diversity was significantly different from that in the fall ($p < 0.017$), while summer community diversity did not significantly differ from community diversity in the spring and autumn. Lastly, the seasonal differences in the Chao1 and ACE diversity indices were similar, where communities in the spring exhibited significantly different diversities compared to those from the summer and autumn
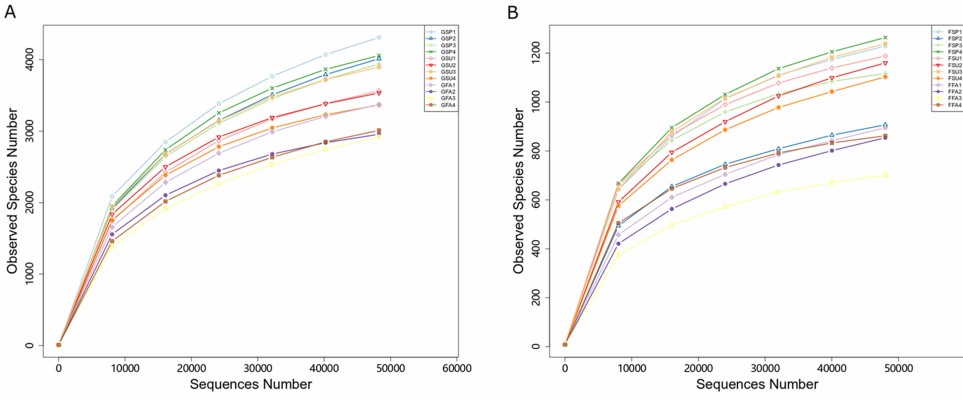

**Figure 3** Rarefaction curves of partial sequences of (A) bacterial 16S rRNA genes and (B) fungal ITS from the elm rhizosphere communities from different seasons. *GSP*, bacterial communities in spring samples; *GSU*, bacterial communities in summer samples; *GFA*, bacterial communities in autumn samples; *FSP*, fungal communities in spring samples; *FSU*, fungal communities in summer samples; *FFA*, fungal communities in autumn samples.

**Table 2** Alpha-diversity indices for bacterial and fungal communities.

| Sample name | Shannon | Simpson | Chao1 | ACE | Good_coverage (%) |
|---|---|---|---|---|---|
| GSP | 9.89 ± 0.09a | 0.997 ± 0.00a | 5097.52 ± 215.68a | 5160.07 ± 164.83a | 97.6 ± 0.00a |
| GSU | 9.58 ± 0.12ab | 0.995 ± 0.00ab | 4046.73 ± 111.24b | 4181.37 ± 153.30b | 98.3 ± 0.00a |
| GFA | 9.08 ± 0.14b | 0.993 ± 0.00b | 3722.48 ± 137.09b | 3820.98 ± 153.22b | 98.3 ± 0.00a |
| FSP | 6.29 ± 0.28a | 0.946 ± 0.02a | 1295.39 ± 85.88a | 1309.66 ± 86.96a | 99.5 ± 0.00a |
| FSU | 6.44 ± 0.07a | 0.962 ± 0.00a | 1406.26 ± 25.92ab | 1421.71 ± 20.92ab | 99.5 ± 0.00a |
| FFA | 5.64 ± 0.32a | 0.937 ± 0.02a | 1017.97 ± 91.24b | 1039.56 ± 87.35b | 99.4 ± 0.00a |

**Notes.**
Different letters within the same parameter indicate significant differences at the $p \leq 0.017$ level (Bonferroni corrected results).
GSP, bacterial communities in spring samples; GSU, bacterial communities in summer samples; GFA, bacterial communities in autumn samples; FSP, fungal communities in spring samples; FSU, fungal communities in summer samples; FFA, fungal communities in autumn samples.

($p < 0.017$), but significant differences were not observed between summer and autumn rhizosphere communities (Table 2). The α-diversity indices for fungal communities all followed trends wherein diversity was highest in summer, followed by diversity in the spring and then autumn (Table 2). One-way ANOVA of Shannon and Simpson indices indicated lack of significant differences among seasons. In contrast, one-way ANOVA indicated significant differences in the Chao1 and ACE indices among the communities from the three seasons. Subsequent multiple pairwise comparisons indicated that autumn community diversity was significantly different from the diversity of the spring ($p < 0.017$), while the community diversity from summer did not significantly differ from the diversity observed in the spring and autumn (Table 2).

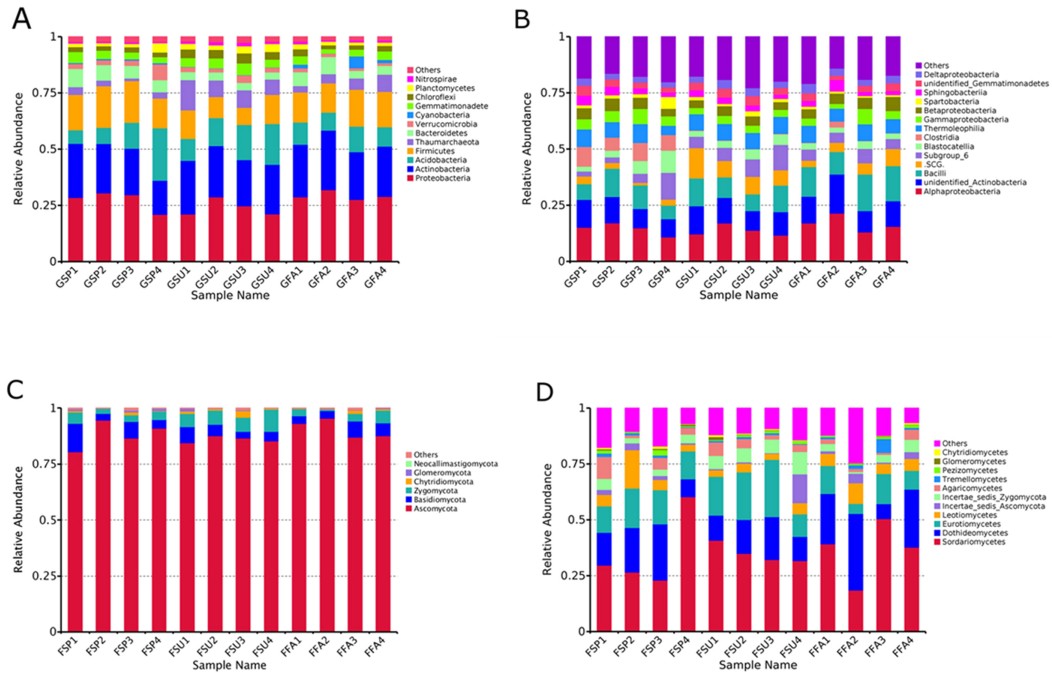

**Figure 4** **Relative abundances of dominant microbial taxonomic groups in elm rhizospheres.** (A) Dominant bacterial phyla. (B) Dominant bacterial classes. (C) Dominant fungal phyla. (D) Dominant fungal classes. Others represent taxonomic groups with low content or those that are unclassified. *GSP*, bacterial communities in spring samples; *GSU*, bacterial communities in summer samples; *GFA*, bacterial communities in autumn samples; *FSP*, fungal communities in spring samples; *FSU*, fungal communities in summer samples; *FFA*, fungal communities in autumn samples.

## Variation in microbial community taxonomic composition of elm rhizospheres among seasons

The rhizosphere-associated bacterial OTUs were associated with 40 total phyla, of which 12 were in >0.6% average relative abundance, and are shown in Fig. 4A. Most of the rhizospheric bacterial sequences (95.1%) belonged to the Proteobacteria, Actinobacteria, Acidobacteria, Firmicutes, Bacteroidetes, Verrucomicrobia, Gemmatimonadete, Chloroflexi, and Planctomycetes. Proteobacteria was the most abundant phylum in nearly all samples, with the exception of GSP4, GSU1, and GSU4. Acidobacteria was the most abundant phylum in GSP4, while Actinobacteria dominated the GSU1 and GSU4 communities. Of the above phyla, Proteobacteria, Actinobacteria, Acidobacteria, and Firmicutes represented the largest components of each rhizosphere soil community, together comprising at least 67% of the total bacterial communities within each sample (Fig. 4A). To assess taxonomic variation among soils at a finer taxonomic resolution, bacterial classes that were in >3% relative abundance of the rhizosphere communities were further analyzed (Fig. 4B). Four classes (Alphaproteobacteria, Gammaproteobacteria, Betaproteobacteria, and Deltaproteobacteria) were identified within the Proteobacteria, two (unidentified Actinobacteria, and Thermoleophilia) from the Actinobacteria, two (Subgroup_6, and Blastocatellia) from the Acidobacteria, two (Bacilli and

Clostridia) from the Firmicutes, one (Spartobacteria) from the Verrucomicrobia, one (Spartobacteria) from the Bacterioidetes, and one (unidentified Gemmatimonadetes) from the Gemmatimonadetes. The most abundant classes among all rhizosphere communities were the Alphaproteobacteria, unidentified Actinobacteria and Bacilli, which together comprised at least 24.93% of the total bacterial communities in each sample. The next most abundant classes were the 'Subgroup_6, of the Acidobacteria, Thermoleophilia, Gammaproteobacteria, and Betaproteobacteria, which comprised at least 16.97% of the total bacterial communities within each sample. Clostridia populations were noticeably more abundant in the spring samples, comprising at least 5.8% of the bacterial communities in these samples (Fig. 4B).

A total of 28 fungal classes were identified in the rhizosphere soils that belonged to six phyla including the Ascomycota, Basidiomycota, Zygomycota, Chytridiomycota, Glomeromycota, and Neocallimastigomycota. All of these phyla were present in every sample, with the exception of the Neocallimastigomycota that were only present in spring and summer samples. The Ascomycota overwhelmingly dominated the communities for all samples, and comprised at least 80.53% of the total fungal communities for every sample (Fig. 4C). To assess taxonomic variation among soils at a finer taxonomic resolution, fungal classes in >0.6% relative abundances in soils were analyzed (Fig. 4D). Six classes (Sordariomycetes, Dothideomycetes, Eurotiomycetes, Leotiomycetes, Incertae_sedis_Ascomycota, and Pezizomycetes) were identified from the Ascomycota, two (Agaricomycetes and Tremellomycetes) from the Basidiomycota, one (Incertae_sedis_Zygomycota) from the Zygomycota, one (Glomeromycetes) from the Glomeromycota, and one (Chytridiomycetes) from the Chytridiomycota. The most abundant classes among all rhizosphere samples were the Sordariomycetes, Dothideomycetes, and Eurotiomycetes, which together comprised at least 52.58% of the total fungal populations in each of the samples. In particular, the classes Leotiomycetes and Incertae_sedis_Zygomycota together comprised at least 6.85% of the total fungal populations in each of the samples (Fig. 4D).

## β-diversity of elm rhizosphere microbial communities among seasons

The NMDS ordination of the elm rhizosphere bacterial communities exhibited a stress value of 0.033 (Fig. 5A), which indicated appropriate representation of community compositional dissimilarities (*Kruskal, 1964*). Bacterial communities segregated by season of origin, wherein NMDS axis 1 separated the spring communities from the fall and summer communities, and NMDS axis 2 separated the fall and summer communities. Overall the bacterial community structure in summer and autumn was relatively similar (Fig. 5A). The stress value of the fungal community NMDS ordination was 0.119, which similarly indicated that the plots were appropriate for representing community dissimilarities (Fig. 5B). As in the bacterial community analyses, the NMDS ordinations indicated a general segregation of fungal communities by season, albeit with less discrete clustering by season. NMDS axis 1 generally separated the spring rhizosphere communities from those in the summer and
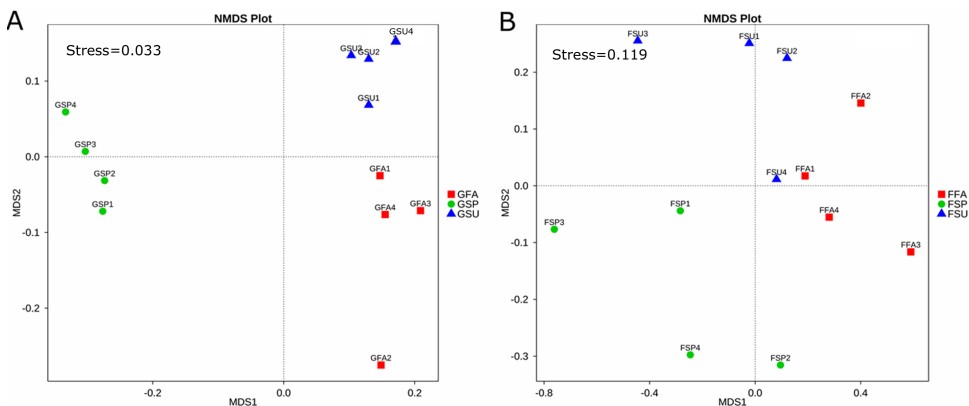

**Figure 5** NMDS analysis of (A) bacterial and (B) fungal community compositions of elm rhizosphere samples taken from three different seasons. *GSP*, bacterial communities in spring samples; *GSU*, bacterial communities in summer samples; *GFA*, bacterial communities in autumn samples; *FSP*, fungal communities in spring samples; *FSU*, fungal communities in summer samples; *FFA*, fungal communities in autumn samples.

**Table 3** PERMANOVA on the effects of seasons on the microbial diversity of the elm rhizosphere soil.

|  | *df* | *F* | *p* |
|---|---|---|---|
| GSU-GFA | 1 | 2.8234 | 0.041* |
| GSU-GSP | 1 | 6.2309 | 0.001** |
| GFA-GSP | 1 | 5.6672 | 0.029* |
| FSU-FFA | 1 | 1.2021 | 0.166 |
| FSU-FSP | 1 | 1.3169 | 0.142 |
| FFA-FSP | 1 | 1.6186 | 0.024* |

**Notes.**
*$p < 0.05$.
**$p < 0.01$.
$p < 0.05$ indicates significant differences between the two groups.
GSP, bacterial communities in spring samples; GSU, bacterial communities in summer samples; GFA, bacterial communities in autumn samples; FSP, fungal communities in spring samples; FSU, fungal communities in summer samples; FFA, fungal communities in autumn samples.

autumn, while axis 2 generally separated communities from the summer and autumn. Overall the fungal community structure in summer and autumn was similar (Fig. 5B).

PERMANOVA tests indicated significant differences among bacterial communities from the summer and autumn samples ($F = 2.8234$, $p = 0.041$), summer and spring samples ($F = 6.2309$, $p = 0.001$), and autumn and spring samples ($F = 5.6672$, $p = 0.029$). Likewise, significant differences in fungal communities were observed only from the autumn and spring samples ($F = 1.6186$, $p = 0.024$) while the summer fungal communities were not significantly different from the spring and autumn communities ($p > 0.05$) (Table 3).

No bacterial taxonomic groups were significantly associated with summer samples and the number of groups associated with spring samples were greater than for autumn samples (Fig. 6). In particular, Bacteroidetes, Firmicutes, and Verrucomicrobia were important taxa that distinguished spring bacterial communities. The Bacteroidia class of the Bacteroidetes group was responsible for their delineation of spring communities.

## Cladogram

**Figure 6   LEfSe analysis of elm rhizosphere microbial communities.** LEfSe analysis was used to identify significantly enriched bacterial taxonomic groups among seasonal samples. In the LEfSe-based cladogram, concentric circles radiating from the inside to the outside represent taxonomic levels from the phylum to the genus (or species) level. Each smaller closed circle at the different taxonomic levels represents one taxonomic unit at this level, and the diameter of the circle is proportional to the taxa's relative abundance. The cladogram shows the taxonomic representation of statistically differences in taxonomic groups among spring, summer, and autumn bacterial communities. Differences are represented in the color of the most abundant class (red: autumn, green: spring, blue: summer, yellow: non-significant). SCG corresponds to a class of the Thaumarchaeota phylum of Archaea.

Likewise, the Ruminococcaceae family of the Firmicutes phylum and Clostridiales order also discriminated spring communities, while the Clostridia, and Lactobacillaceae family of the Lactobacillales order also distinguished spring communities. In contrast, the Bacillaceae and Planococcaceae families of the Bacillales order differentiated autumn samples from others (Fig. 6).

## Effects of soil physicochemical properties on elm rhizosphere microbiota

To evaluate the environmental factors influencing bacterial and fungal community structures of elm rhizospheres, redundancy discriminant analysis (RDA) was used to assess community-environment relationships. The RDA analyses indicated that different environmental factors exhibited different effects on bacterial and fungal communities, wherein the main influencing factors also varied among seasons (Fig. 7). The first axis of the bacterial biplot explained 30.4% of the variation in sample-environment relationships, while the first and second axes together explained 50.9%. MC was the most significantly correlated environmental variable to bacterial community composition, while AK was also significantly correlated. The environmental factors that primarily affected bacterial

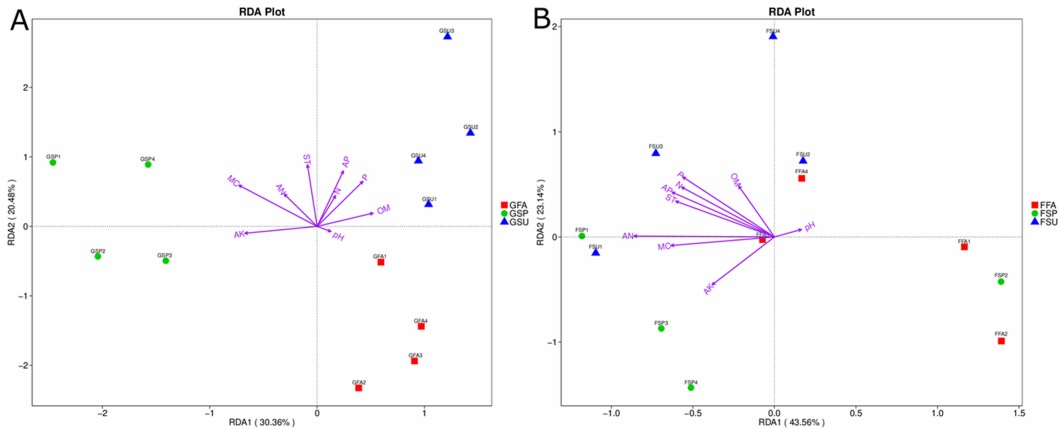

**Figure 7  Redundancy discriminant analysis (RDA) biplot showing relationships between (A) bacterial or (B) fungal communities and environmental variables in elm rhizospheres.** The distances between samples in the RDA biplot indicate the affinity-disaffinity relationships of community structures. The length of the arrow is proportional to the degree of influence from environmental variables on community structures. Further, the closer the sample point is to a given environmental variable, the greater the influence of the environmental variable on that sample. *N*, total N; *AN*, available N; *P*, total P, *AP*, available P; *AK*, available K; *OM*, organic matter; *MC*, moisture content; *ST*, soil temperature. *GSP*, bacterial communities in spring samples; *GSU*, bacterial communities in summer samples; *GFA*, bacterial communities in autumn samples; *FSP*, fungal communities in spring samples; *FSU*, fungal communities in summer samples; *FFA*, fungal communities in autumn samples.

communities in the spring were MC and AK, while the factors that affected bacterial communities in the summer were primarily P, AP, and OM. Lastly, pH exhibited a minor correlation to bacterial community composition in autumn samples (Fig. 7A). For fungal community variation, the first axis explained 43.6% of the variation in sample-environment, relationships, while the first and second axes together explained 66.7% of the variation. AN was the environmental variable mostly explaining fungal community compositions, although P, AP, N, and ST were also influential. Lastly, pH was modestly influential towards community compositional variation (Fig. 7B).

## Bacterial community functional prediction

A total of 41 level two KEGG Orthology (KO) groups were identified in the bacterial communities that were distributed across six metabolic pathways. Among these pathways, those involved in metabolism, genetic information processing, and environmental information processing were most prevalent, accounting for 52.15%, 15.7%, and 13.65% of the totals. At a finer level of resolution, the inferred relative abundances of gene families associated with membrane transport (11.41%), amino acid metabolism (11.14%), and carbohydrate metabolism (10.62%) were particularly high in the elm rhizosphere communities (Fig. 8). Nevertheless, multiple comparison analyses indicated that the predicted distribution of gene functions did not significantly differ among rhizosphere communities from different seasons ($p > 0.05$).

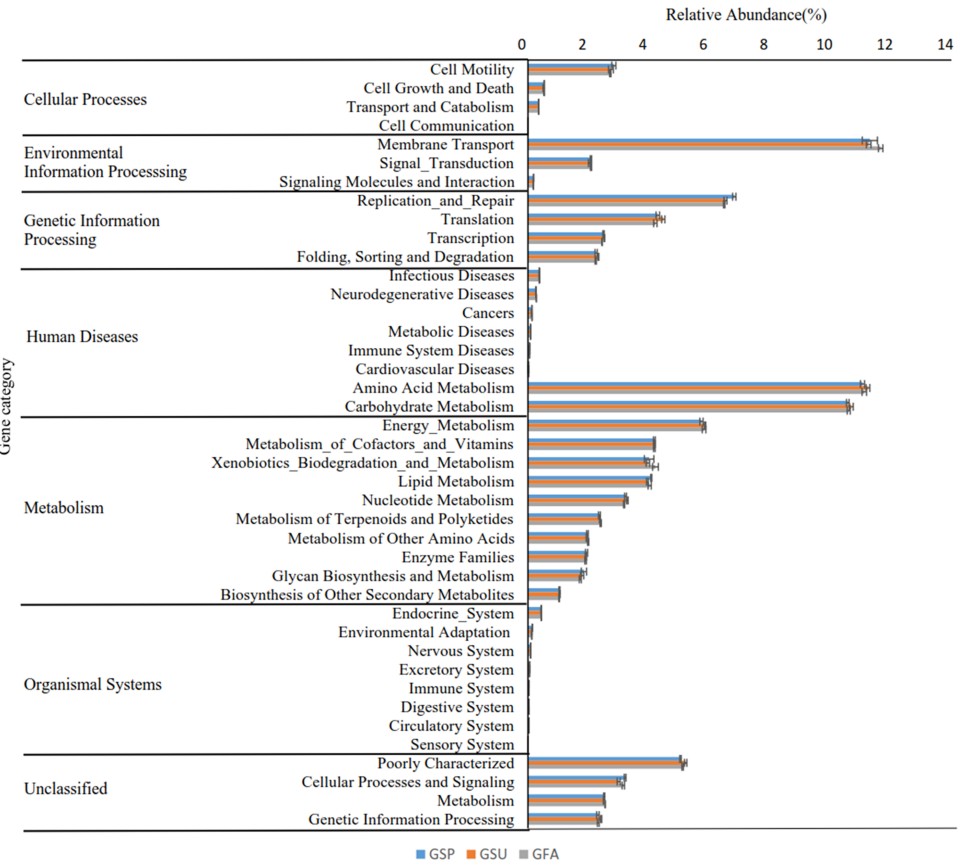

**Figure 8  Predicted functions of the bacterial communities found in rhizosphere of the elm.** Bonfferoni correction for multiple comparison indicates there are no significance among three seasons ($p > 0.05$). *GSP*, bacterial communities in spring samples; *GSU*, bacterial communities in summer samples; *GFA*, bacterial communities in autumn samples.

## Fungal community functional analyses

Functional analysis of the fungal communities using the FUNGuild software program indicated that eight different ecological guilds were present among the fungal communities of the elm rhizospheres, in addition to unidentified guilds. Among these, saprotrophs and pathotrophs were most abundant, accounting for 32.1% and 11.1% of the total communities (Fig. 9). Multiple comparison analyses indicated that the distribution of fungal functional guilds did not significantly differ among rhizosphere communities from different seasons, as was observed for the bacterial communities ($p > 0.05$).

## DISCUSSION

Soil microorganisms play critical roles in the flow of energy and materials in ecosystems (*Martina Štursová et al., 2012*). In particular, rhizosphere microorganisms are essential for plant growth, alleviate pathogen colonization, and maintain rhizosphere microecological balance (*Li et al., 2014*; *Zhou et al., 2014*), and it is increasingly recognized that harnessing rhizospheric microbiomes offers new opportunities to influence plant performance

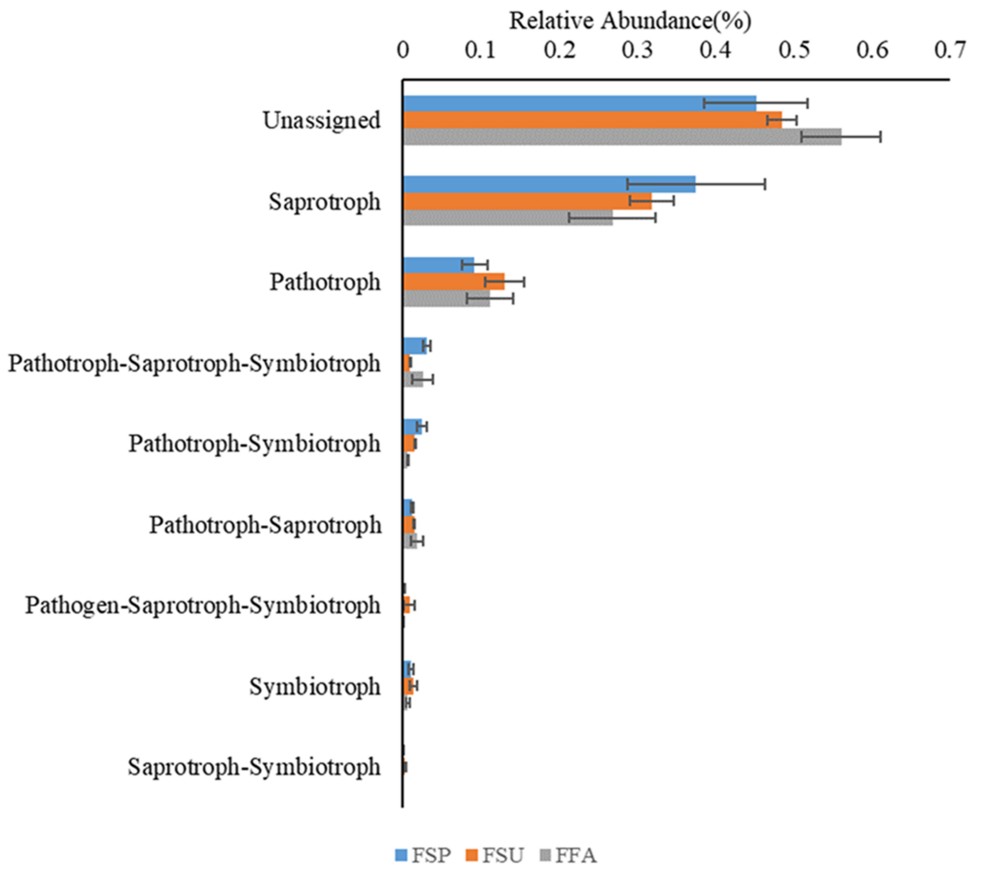

**Figure 9 Ecological guilds of the fungal communities found in rhizosphere soil of the elm.** Bonferroni correction for multiple comparison indicates there are no significance among three seasons ($p > 0.05$). *FSP*, fungal communities in spring samples; *FSU*, fungal communities in summer samples; *FFA*: fungal communities in autumn samples.

through adequate management (*Sutherland et al., 2019*). The present pilot study used high-throughput sequencing of community phylogenetic marker genes to evaluate the microbial diversity in *Ulmus pumila* L. var. *sabulosa* rhizospheres in the Otindag Sandy Land across three seasons. To our knowledge, this is the first high-throughput sequencing analysis of bacterial and fungal diversity of *Ulmus pumila* L. var. *sabulosa* rhizospheres. Nearly all of the bacterial and fungal diversity within the rhizosphere samples were adequately sampled using our high-throughput sequencing approach (Fig. 3). However, we acknowledge the limited sampling scope of the study, and it cannot be absolutely determined that the sampling of the OSL hinterland was representative of the ecosystem diversity across this important area. Nevertheless, these results provide new insights into the seasonal dynamics of rhizosphere microbial community structures of *Ulmus pumila* L. var. *sabulosa* in the OSL hinterland.

Given the extensive richness of OTUs, quantitative comparisons of taxonomic differences were only conducted at the higher taxonomic levels, as previously described (*Peiffer et al., 2013*). Proteobacteria, Actinobacteria, Acidobacteria, and Firmicutes comprised the largest

components of each rhizosphere soil community. It is possible that these observations may be due to primer amplification biases. A survey of 16S rRNA gene primer coverage indicated that the primers used here exhibited high coverage of the Proteobacteria, Actinobacteria, Acidobacteria, and Firmicutes (Table S4 ). Proteobacteria were clearly the dominant phylum of elm rhizospheres in the OSL, which is consistent with investigations of rhizosphere communities of other trees. For example, Proteobacteria were the primary bacterial taxa observed in *Populus deltoids* rhizosphere soils (*Gottel et al., 2011*) and also those of *Pinus tabulaeformis* (*Yu, Wang & Tang, 2013*). *Spain, Krumholz & Elshahed (2009)* suggested that Proteobacteria was the most common phylum in soils globally because they generally grow fast like weedy species and are well known to rapidly respond to unstable carbon sources. It should however be noted that Actinobacteria, rather than Proteobacteria, were the dominant taxa in the GSU1 and GSU4 samples (Fig. 2A). Actinobacteria have been observed to exhibit uniquely increased survival rates through periods of environmental stress (*Leggett et al., 2012*). Similar observations have been reported in non-disturbed grasslands (*Yu et al., 2011*). Thus, similar mechanisms may explain the dominance of Actinobacteria in these two samples. The Ascomycota were by far the most abundant phylum in the elm rhizosphere communities, which is consistent with other investigations of rhizospheres from trees and grasslands. For example, Ascomycota dominated the rhizosphere communities of *Taxus* trees (*Hao et al., 2016*) and were also the predominant taxa in tropical grasslands of Laos (*Lienhard et al., 2014*).

Seasonal variation in the α-diversity of bacterial and fungal communities in elm rhizospheres was evaluated using the Shannon (*Shannon & Weiner, 1949*) and Simpson (*Simpson, 1949*) diversity indices that encompass species richness and evenness metrics. Further, diversity was also evaluated using the Chao1 (*Chao, 1984*) and ACE (*Chao & Lee, 1992*) estimator indices that reflect estimated community richness. Variation in bacterial α-diversity was significantly correlated to season, indicating that seasonal changes in soil environments significantly impact the richness and evenness of elm rhizosphere bacterial communities. However, seasonality only significantly influenced the Chao1 and ACE richness indices of fungal communities. Consequently, season-induced changes in soil environments may only significantly influence the richness of elm rhizosphere fungal communities, but not their evenness (Table 2). These results are consistent with investigations of *Pinus roxburghii* rhizosphere communities, wherein season significantly influenced the richness and evenness of bacterial and fungal communities within those rhizospheres (*Yadav, 2013*). α-diversity values for bacterial and fungal communities were higher in the earlier growing season, and decreased in fall and summer (Table 2). The gradual rise of soil temperature in May leads to plant germination and growth of vegetation could provide a range of resources like exudation for microbes to maintain a higher diversity during this time (*López-Mondéjar et al., 2015*). Between May and August, gradually increasing precipitation levels leads to soil hardening and decreased soil aeration and water infiltration. Thus, anaerobic conditions may restrict the growth of aerobic microorganisms, thereby leading to declining diversity in summer (*Li et al., 2012*).

Variation in β-diversity patterns reflected the seasonal influences on microbial diversity within elm rhizospheres. In particular, bacterial community structures significantly

differed among seasons, while fungal community structures only significantly differed between spring and autumn (Table 3). As with bacterial α-diversity, bacterial community β-diversity in elm rhizosphere soils was more responsive to changes in soil environments than were fungal communities. Several studies have shown that fungi have greater tolerance to environmental changes than bacteria, and that fungi have lower requirements for water and nutrients. Consequently, fungi can grow well in environments with poor resources, and do not grow especially well in environments with sufficient water and nutrients (*Yuste et al., 2011*; *Patra, 1990*). Differing ecological strategies based on responsiveness to environmental changes between bacteria and fungi underlie these different adaptive mechanisms. Bacteria exhibit typical r-strategies. That is, bacterial cells continuously uptake nutrients from the environment to accelerate growth and reproduction when water content and nutrients are high. Conversely, insufficient nutrient availabilities lead to rapid decreases in bacterial populations and low recovery rates. In contrast, fungi exhibit k-strategies, wherein populations are relatively stable. When populations decline, populations can rapidly restore as long as environmental changes do not exceed adaptive ranges (*Deyn et al., 2011*). Overall, these results indicate that elm rhizosphere microbial diversity responds to seasonal environmental fluctuations. However, functional analysis indicated that season did not significantly influence inferred bacterial metabolic functions and fungal ecological functions, in contrast to patterns in diversity. It should however be noted that functional gene distributions provide glimpses into the metabolic potential and ecological functions of communities, but do not reflect the actual metabolic activities and ecological functions of communities (*Castañeda & Barbosa, 2017*). Indeed, several studied have indicated significant changes in taxonomic distributions among communities, but comparable functional gene distributions (*Hollister et al., 2010*; *Ossola et al., 2016*). Consequently, gene expression and regulation are integral components to consider when evaluating the link between gene content and metabolic activities.

It is extensively documented that microbial diversity is generally sensitive to environmental changes. Microorganisms respond to environmental changes by modulating gene expression and protein translation, thereby leading to changes in microbial physiological activities and ultimately, differences in population abundances. Hence, microbial community structures change in response to environmentally-induced gene regulation processes. Nevertheless, the presence of functional genes allow great potential for community resilience and the maintenance of community functions in addition to the effects of genetic regulation. For example, the same strains respond differently under different environmental settings. Additionally, low abundance microorganisms may become more dominant under certain conditions, although they largely retain the same functional genes, but exhibit differential gene expressions (*Torsvik & Ovreås, 2002*; *Kaschuk, Alberton & Hungria, 2010*; *Souza et al., 2014*). Variation in soil temperature, humidity, pH, and the level of organic and inorganic nutrients, among other factors that are associated with different seasons, can all affect the structures of microbial communities (*Thoms & Gleixner, 2013*). However, a consensus of the most influential factors has yet to be reached. The results reported here indicate that soil moisture content and available K were key factors that varied with season and affected bacterial communities in elm rhizospheres (Fig. 6A). These

observations coincide with results from other studies. *Rasche et al. (2011)* observed that bacterial communities were tightly coupled to seasonal changes in soil moisture. Further, *Tian & Gao (2014)* observed that available K was the primary factor that affected cucumber rhizosphere bacterial communities. In contrast to bacterial communities, soil nutrients including available N and total P were the primary factors that affected rhizosphere fungal community composition within elm rhizospheres over seasons (Fig. 6B). Investigation of rhizosphere microbial communities for two plants in the Hobq Sandy Land of Inner Mongolia in China also indicated that soil nutrients significantly influenced microbial population numbers (*Dai et al., 2016*). The environmental characteristics of the Hobq Sandy Land are similar to those of the research area investigated here, and the results of the aforementioned study support those of the present analyses. Likewise, soil characteristics were significant in shaping the bacterial and fungal communities of the black soil zone in northwestern China soils (*Liu et al., 2014*; *Liu et al., 2015*). Many studies have shown that soil pH can greatly influence microbial community structure (*Rousk et al., 2010*; *Nacke et al., 2011*). However, pH had little effect on the microbial communities of elm rhizospheres in this study, which may be due to the relatively narrow range of pH for the soils sampled here.

## CONCLUSIONS

In summary, high throughput sequencing of phylogenetic marker genes for bacterial and fungal communities revealed the influence of seasonal variation on the microbial communities associated with *Ulmus pumila* L. var. *sabulosa* rhizospheres. Proteobacteria was the most abundant phylum of rhizosphere bacterial communities, while the Ascomycota fungal phylum was most abundant within fungal communities. Our results indicate that seasonal changes affected the diversity of elm rhizosphere bacterial and fungal communities. Bacterial community diversity was highest in spring rhizospheres, while that of fungal communities was highest in summer rhizospheres. In addition, bacterial community structure significantly differed among seasons, while those of fungal communities only significantly differed between spring and autumn. Microbial community functional analysis indicated that season did not appear to significantly influence the variation in bacterial and fungal functions among elm rhizospheres. Soil moisture content and available K were the most influential factors affecting bacterial community structure among all physiochemical parameters that were measured, while total N and P contents primarily affected fungal community structures. The present study provides important data to understand the responses of elm rhizosphere microbial communities towards seasonal changes in the hinterland of the OSL. Our data also provide a baseline framework for subsequent studies to understand the relationship between elm rhizosphere communities and plants performance which could be useful for understanding the ecological functioning and characteristics of the OSL hinterland. However, subsequent studies should be conducted to evaluate the functions of these rhizosphere communities and the associated interaction mechanisms between rhizosphere microorganisms and elm trees.

## ACKNOWLEDGEMENTS

We appreciate the help from Mr Xiaofu Cheng for field sampling. We would like to thank LetPub for providing linguistic assistance during the preparation of this manuscript.

### Funding

This work was supported by the National Natural Science Foundation of China (No. 31260202) and Forestry Public Welfare Industry Special Scientific Research of China (No. 201504412). The funders had no role in study design, data collection and analysis, decision to publish, or preparation of the manuscript.

### Grant Disclosures

The following grant information was disclosed by the authors:
National Natural Science Foundation of China: 31260202.
Forestry Public Welfare Industry Special Scientific Research of China: 201504412.

### Competing Interests

The authors declare there are no competing interests.

### Author Contributions

- Tianyu Liang conceived and designed the experiments, performed the experiments, analyzed the data, contributed reagents/materials/analysis tools, prepared figures and/or tables, authored or reviewed drafts of the paper, approved the final draft.
- Guang Yang conceived and designed the experiments, analyzed the data, contributed reagents/materials/analysis tools, prepared figures and/or tables, authored or reviewed drafts of the paper, approved the final draft.
- Yunxia Ma conceived and designed the experiments, performed the experiments, prepared figures and/or tables.
- Qingzhi Yao conceived and designed the experiments, contributed reagents/materials/-analysis tools, authored or reviewed drafts of the paper.
- Yuan Ma performed the experiments, prepared figures and/or tables.
- Hui Ma and Yang Hu performed the experiments.
- Ying Yang analyzed the data.
- Shaoxiong Wang and Yiyong Pan contributed reagents/materials/analysis tools, field Sampling.
- Gangtie Li conceived and designed the experiments, authored or reviewed drafts of the paper, approved the final draft.

### Field Study Permissions

The following information was supplied relating to field study approvals (i.e., approving body and any reference numbers):

Our studies was sampled on a private pasture. Field experiments were approved by its owner Ying Tao.

## DNA Deposition

The following information was supplied regarding the deposition of DNA sequences:

The elm rhizosphere soil microbia sequences are accessible via SRA: SRR8784776–SRR8784799.

## Data Availability

The raw measurements are available in the Supplemental Files. The seasonal dynamics of microbial diversity in the rhizosphere of Ulmus pumila L. var. sabulosa were analyzed based on these original data.

## Supplemental Information

Supplemental information for this article can be found online at http://dx.doi.org/10.7717/peerj.7526#supplemental-information.

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
