# Peer review of "Seasonal dynamics of microbial diversity in the rhizosphere of Ulmus pumila L. var. sabulosa in a steppe desert area of Northern China"

_PeerJ, doi:10.7717/peerj.7526_

## Round 0.1 · original submission · Major Revisions

Dear Tian Yu,
Your manuscript has been reviewed by two reviewers, who found that your work has merit but also suffers from some deficiencies. Upon careful consideration, I am ready to consider to evaluate a revised version that would take into account seriously the comments and suggestions they have made.
In particular, please carefully address the issues linked to the -rather low- number of replicates used in this study, which can restrict the capacity to generalize the conclusions and to translate them into management guidelines (i.e. better to recognize the limitations linked to the experimental design).
More generally, take into account all the points raised and suggestions made by the reviewers. Note that one reviewer urged you to adopt the correction for multiple comparisons (likely with Bonferroni correction ; see https://en.wikipedia.org/wiki/Bonferroni_correction).
I look forward to receiving a revised version of your manuscript along with a point by point response to the reviewers' comments and suggestions.
best regards
Xavier

·

Basic reporting

See general comments

Experimental design

See general comments

Validity of the findings

See general comments

Additional comments

The article by Liang et al. investigates the seasonal dynamics of the microbial communities (fungi and bacteria) found in the rhizosphere of Ulmus pumila.

The article is interesting because this tree species, which inhabits arid areas of northeast Asia, plays a major role in reducing wind erosion, accelerating vegetation restoration, and increasing nutrient accumulation.

However, the study has several important limitations:

Your introduction needs to focus on what is known on the rhizosphere of other tree species and less on the ecosystem under study (L42-59). Also, the section about other methodologies is unnecessary (L97-109).

You did not include control (bulk soil) samples, which is a common practice in rhizosphere studies. Why? Without control samples you do not know which OTUs are overrepresented in the rhizosphere compared to the bulk soil.
You stated that the sampling was representative (L404). I do not agree with this. The number of samples (n=12) is quite low. Also, you collected all the samples in one site and soil is spatially very heterogeneous. Including more sites will make the study more representative and robust.

In your material and methods you should include information about ANOSIM and RDA, as they are mentioned later in the manuscript.

You should clarify the analysis of the sequences. What do you mean by “clean tags” and “effective tags”? How many sequences were finally used? Did you rarefy your data? Most studies use PERMANOVA instead of AMOVA. Did you test all pairwise comparison in you beta-diversity analysis (L348-354)?

Quite often, the discussion is a repetition of the results. The discussion should relate the findings of your study to previous studies. That is, you should contextualize the contribution of your study. For example L420-424.

I do not see how “these results provide valuable context to inform the ecological restoration of this area, and provide significant data to evaluate the ecological functions of common and rare microorganisms in such ecosystems in future studies” (L496-499). I would say that the low number of samples and the sampling design make this almost impossible.

Minor comments:

Did you use 18S rRNA genes (L113) or ITS (L23) for the study of fungal communities? By the way, the ITS is not a gene but a region between the 16S and 23S rRNA genes.

L357-360. “In the LEfSe … relative abundance”. This information should be in a figure legend and not in the results section.

L398-401. “A total of … respectively”. This information should be part of the results section.

L426. You mention that you sequencing depth was quite limited, but earlier (L400-401) you stated than the coverage was high? This seems a bit contradictory, isn’t it?

·

Basic reporting

Literature references:
Referencing should be checked as for e.g. mothur, wrong reference was supplied:
line 234 : ... within Mothur (Kropf, Ulrich, Tebbe, 2007).

Experimental design

Materials and methods:
Please clarify the section around line # 43 ... Thus, a total of 12 soil samples were collected among the three seasons.... Comment on the size (mass) of the samples collected. How representative were these samples? Have the authors additional information that supports the notion that such an approach is sufficiently robust for the type of analyses they performed?

Please consider using correction for multiple comparisons in significance tests such as Benjamini-Yekutieli, or Benjamini-Yekutieli-Hochberg.

With respect to data usage, I would also suggest to extend the analyses of bacterial 16S rRNA genes using functional imputation (programs Piecrust, Tax4Fun, or other) that would shed the light into functional gene distribution of these soils and provide an additional insight. This could be accomplished just by including a short section within Electronic Supplementary Materials.

Validity of the findings

The authors are urged to assess their significance thresholds by adopting the correction for multiple comparisons.

Conclusions are otherwise well suited.

Additional comments

Please check Figure caption as some mistakes were found.
Figure 4 -> stress values are hardly seen
Figure 6 -> legend is corrupt and the lines are broken


Figure captions -> consider extending the description of the names of your samples in figure captions:
GSP: bacterial communities in spring samples; GSU: bacterial communities in
summer samples; GFA: bacterial communities in autumn samples; FSP: fungal communities in
spring samples; FSU: fungal communities in summer samples; FFA: fungal communities in autumn samples.

---

## Round 0.2 · Major Revisions

dear authors,

I appreciate the efforts you made during the first round of revision of this ms. However, one reviewer considered that you did not adequately addressed his/her concerns. This is particularly true regarding (i) the way you describe the sampling design (see lines 249-257), and (ii) your claiming that your results are representative of this whole ecosystem. I have worked a bit on your ms. and proposed a way to better present, with sufficient details, your sampling design. in addition, I have suggested ways to avoid claiming too much in term of representativeness. More generally, I have made a range of suggestions in the manuscript; please feel free to accept them or not (just explain why when you disagree).
I will be pleased to consider a version taking into account these last feedbacks. I hope that you understand that the objective remains to improve the quality of your ms. before we can consider publishing it.
Note: please revise the table/figure captions, it seems that you have typos in them (missing spaces...); also, please check the homogeneity of the format of references (points/no points to authors' initials, etc.)
Best regards
Xavier

·

Basic reporting

See below

Experimental design

See below

Validity of the findings

See below

Additional comments

Most of my concerns have not been properly addressed.

·

Basic reporting

The revised (R1) version of this manuscript is a significant improvement over the originally submitted manuscript.

Some references are still not correct, e.g. the correct reference for mothur program is
https://aem.asm.org/content/75/23/7537
Introducing mothur: Open-Source, Platform-Independent, Community-Supported Software for Describing and Comparing Microbial Communities
DOI: 10.1128/AEM.01541-09

Do correct this and other references.

Experimental design

The number of samples is rather low, however authors were urged to put this low number of samples into a wider context.
The provided explanations described within their rebuttal letter could form a part supplementary material, to clarify this issue sufficiently.

Otherwise the research question was well defined and fits the scope of the Journal.

Validity of the findings

Authors improved significantly their analyses, included permanova and functional gene prediction set, provided correction for multiple comparisons.
Once the background explanation (see#2. Experimental design) is provided and included as part of the ESM, the findings will be easier to contextualize.

The findings are valid, but need to be placed within the wider scope due to characteristics of the local environment where sampling was performed.

Additional comments

Correct the references again.

Discussion section on Predicted functional genes ln 559-573:
However, inferred metabolic functional analysis indicated that season did not significantly influence inferred bacterial metabolic functions, unlike the results for diversity. Nevertheless, these analyses still indicated rich bacterial functional diversity in these rhizospheres, with the detection of 41 level two KO groups. It should be noted that there are considerable limitations in the inference of community metabolic functions via the PICRUSt analytical approach. Consequently, further studies are needed to evaluate the functions of elm rhizosphere microorganisms using other analytical tools like metagenome sequencing.

Please rephrase this section:
Functional gene distribution gives clues about metabolic potential, not the exact metabolic functions, metabolic activities.
There have been a number of reports where significant changes have been found at the level of taxonomic descriptions, but metagenomic analyses showed basically rather comparable functional gene distribution, giving rise to the conclusion, that it is the management of this genetic potential, i.e. the gene expression and regulation, that generates the observed functional differences at the ecosystem level -> nutrients for plants, plant growth, plant health, ...

Variation in soil temperature, humidity, pH, and the level of organic and inorganic nutrients, among other factors that are associated with different seasons, can all affect the structures of microbial communities (Thoms, Gleixner, 2013). water and K content were important in this study.

environment changes -> microbes respond with adjustment in expression of their genetic material and enzymes produced -> changes in microbial physiology, activities ->some taxa grow, some don't, some die -> hence the change in microbial community structure -> yet the functional gene distribution may well remain the same, except for the fine-tuning elements of genetic regulation; the same microbes behave differently under different environmental settings; microbes from the tail of their abundance distribution may become more dominant, but they largely contain the same functional genes, just express them differently.
For environmental settings there is a need to move away from the classical medical model of one/few microbe/s one disease/effect.

This is the part that I feel would need to be included as part of the manuscript, as this gives an extended view of the importance of the presented results.

---

## Round 0.3 · Minor Revisions

dear Authors,

Two reviewers have now evaluated the revised version of your ms.

After careful evaluation, and despite the divergence of appreciation between the two reviewers, I have decided that this manuscript would be acceptable for publication, provided you take well into account the suggestions made by the reviewer 3.

best regards

xavier LE ROUX

·

Basic reporting

See below

Experimental design

See below

Validity of the findings

See below

Additional comments

In my opinion, there is not real improvement in the revised version of the manuscript.

·

Basic reporting

I would suggest to soften the claim in line 87-88, as the cited review only reports 'up to 30.000' species, not that the rhizosphere generally contains that many species. Also I would suggest citing the primary research article (Mendes et al. 2011) instead of the review.

The language in the two paragraphs from line 481 to 538 needs to be improved, as some sentences are hard to parse in the current form (e.g. line 506-508).

Experimental design

I am not sure it is clear from the manuscript if the same 3 trees per sampling location were sampled for the 3 sampling times or if 3 different trees were sampled each time. Please explain this more clearly.

I think that a survey of primer coverage (e.g https://www.arb-silva.de/search/testprime/ for 16S) would help in discussing why certain taxonomic groups might be missing or could be proportionally more abundant than others.

I would be interested to see an ecological/functional characterization of the fungal communities as well, by using a tool such as FUNGuild (https://github.com/UMNFuN/FUNGuild) or simply discussing the ecological roles of Fungi detected here (i.e. are they saprophytes or symbionts etc.). In my opinion that could improve the coherence of the manuscript, but might not be crucial either.

Validity of the findings

In table 3 (cited in line 483-484), please provide the values for the fungal communities, even if they are not significant.

---

## Round 0.4 · accepted · Accept

Thank you for this final revision of your ms., which is now suitable for publication.

I have detected some typos in the revised parts in particular, which I will forward to the journal staff.

Congratulations
Xavier